# Significantly Improve the Thermal Conductivity and Dielectric Performance of Epoxy Composite by Introducing Boron Nitride and POSS

**DOI:** 10.3390/nano14020205

**Published:** 2024-01-17

**Authors:** Hongnian Long, Wenlong Liao, Rui Liu, Ruichi Zeng, Qihan Li, Lihua Zhao

**Affiliations:** 1College of Electrical Engineering, Sichuan University, Chengdu 610065, China; 2Electric Power Research Institute, State Grid Corporation of Sichuan Province, Chengdu 610072, China; 3College of Aviation Engineering, Civil Aviation Flight University of China, Guanghan 618307, China

**Keywords:** boron nitride nanosheets, polyhedral oligomeric silsesquioxane, nanocomposites, low dielectric constant, high thermal conductivity

## Abstract

Dielectric materials with superb thermal and electrical properties are highly desired for high-voltage electrical equipment and advanced electronics. Here, we propose a novel strategy to improve the performance of epoxy composites by employing boron nitride nanosheets (BNNSs) and γ-glycidyl ether oxypropyl sesimoxane (G-POSS) as functional fillers. The resultant ternary epoxy composites exhibit high electrical resistivity (1.63 × 10^13^ Ω·cm) and low dielectric loss (<0.01) due to the ultra-low dielectric constants of cage-structure of G-POSS. In addition, a high thermal conductivity of 0.3969 W·m^−1^·K^−1^ is achieved for the epoxy composites, which is 114.66% higher than that of pure epoxy resin. This can be attributed to the high aspect ratio and excellent thermally conductive characteristics of BNNSs, promoting phonon propagation in the composites. Moreover, the epoxy composite simultaneously possesses remarkable dynamic mechanical properties and thermal stability. It is believed that this work provides a universal strategy for designing and fabricating multifunctional composites using a combination of different functional fillers.

## 1. Introduction

Epoxy resins are widely utilized in high-voltage insulating equipment and power electronics such as generators, transformers, high-voltage insulating bushings, and cable joints in terms of their toughness, electric insulation properties, as well as being easy to process [1,2,3,4,5,6]. With the gradual development of this equipment towards high-voltage, high power, miniaturization, and integration, further requirements are put forward for the performance of materials. In particular, the thermal management performance of the material presents a severe challenge because a lot of heat will be generated in those application scenarios. However, due to the fact that epoxy resins are formed through the interwinding and crosslinking of irregular polymer molecular segments, their intrinsic thermal conductivity is very low. Such shortcomings may cause heat to accumulate rapidly in the equipment, compromising its service life and performance. [7,8,9,10,11].

It is well known that filling thermally conductive filler into epoxy resin is an effective method to improve the thermal conductivity of composites [12,13,14]. Boron nitride nanosheets (BNNSs), as a graphene-like two-dimensional nanomaterial, possess outstanding thermal conductivity and are an ideal filler for improving the thermal conductivity of epoxy composites [7,15,16,17]. For example, Han et al. prepared thermally conductive epoxy composites by introducing a hybrid filler of SiC nanoparticles and BNNSs. A high thermal conductivity of 0.89 W·m^−1^·K^−1^ has been achieved for epoxy composites while the filler contents are 20 wt%, which is 4.1 times that of pure epoxy resin [18]. Zhao and coworkers reported that an efficient three-dimensional thermal conductive network can be successfully built with two-dimensional BNNSs and zero-dimensional spherical boron nitride as hybrid functional fillers. As a result, the thermal conductivity of composites reaches a high value of 1.148 W·m^−1^·K^−1^ with the fillers loading of 30% [19].

In addition to high thermal conductivity, excellent dielectric properties are also important for many applications. For instance, in some coil equipment such as switch cabinets, reactors, and transformers, epoxy resin is also often used as insulation materials, where low dielectric constant and dielectric loss are required to achieve excellent performance and service life of the equipment. Unfortunately, relevant studies have shown that the addition of BNNSs to epoxy resin will lead to an increase in the dielectric constant of the resultant composite because of the high dielectric constant of the BNNS and the existence of plentiful interface polarization [20,21].

Therefore, it is necessary to introduce a nanomaterial that can remedy the increased dielectric constant brought by the addition of BNNSs to an ideal range. The introduction of polyhedral oligomeric silsesquioxane (POSS) nanofillers has been recognized as an effective method to optimize the dielectric performance of polymer composites [22,23,24,25]. For instance, Lee et al. prepared a new hybrid system using a combination of polyimide and POSS. When the POSS content was 10 wt%, the dielectric constant of the composites was as low as 2.65 [25]. Chen et al. synthesized a composite using POSS and meta-phenylenediamine. Due to the existence of nanoporous POSS cubes in the epoxy group of the composite, its dielectric constant was only 2.31 [26].

The performance improvement of epoxy resin composites using a single filler is very limited. In a large number of studies nowadays, a variety of filler combinations are adopted to improve the thermal conductivity and dielectric properties of epoxy resin at the same time [20,27,28,29]. In the study of T. Heid et al., after 1 wt% of POSS and 5 wt% of cubic boron nitride were mixed into epoxy resin, the dielectric constant and dielectric loss of the composite decreased significantly compared with that of pure epoxy resin. Meanwhile, its thermal conductivity reached 125% that of pure epoxy resin [28]. Wu et al. reported that epoxy composites containing 5 wt% of POSS and 20 wt% of boron nitrides were prepared, which kept the dielectric constant and dielectric loss within a low range. At the same time, the thermal conductivity of the nanocomposite material reached 1.28 W·m^−1^·K^−1^, 6 times that of the pure epoxy resin [20].

In this study, γ-glycidyl ether oxypropyl sesimoxane (G-POSS) and BNNSs have been selected as functional fillers to optimize the thermally conductive and dielectric performance of epoxy composites. G-POSS was specifically chosen because it has a cage structure that can reduce the dielectric constant of the epoxy, and the γ-glycidyl ether groups can also form strong covalent bonds with the epoxy matrix. A series of G-POSS/BNNS/epoxy composites with different mass gradients were prepared by blending G-POSS and BNNS with a certain proportion of filler and filling them evenly into epoxy resin through high-speed stirring and ultrasonic treatment. To evaluate the performance improvement of the epoxy resin using different fillers, the thermal conductivity and dielectric properties of the composites were systematically studied. Finally, we located the recipe to improve the thermal conductivity of the composite significantly and maintain the permittivity and dielectric loss at a low level. The resultant G-POSS/BNNS/epoxy composites have both high thermal conductivity and excellent dielectric properties.

## 2. Experimental Section

### 2.1. Materials

The average particle size of the h-BN powder used in the experiment was 10 μm, which was purchased from China Qinhuangdao Yinuo High-tech Materials Development Co., LTD. (Qinhuangdao, China). The epoxy compound used was bisphenol type A diglycidyl ether (E-51 epoxy resin), equivalent weight 192, purchased from Nantong Xingchen Synthetic Materials Co., Ltd. (Nantong, China). The curing agent was methyl tetrahydrophthalic anhydride (MTHPA), which was purchased from China Jinghong Polymer Materials Co., Ltd. (Jinghong, China). The accelerator was three (dimethylamine methyl) phenol, model DMP-30, produced by Changzhou Runxiang Chemical Co., Ltd. (Changzhou, China). The performance of the epoxy resin was optimized with G-POSS, which was γ-glycidyl ether oxysilpropyl sesioxane purchased from Shanghai Aladdin Co., Ltd. (Shanghai, China). Acetone and trisodium citrate dihydrate were purchased from Chengdu Cologne Chemical Co., Ltd. (Chengdu, China). Other solvent materials in this experiment, including deionized water and anhydrous ethanol, came from Chengdu Cologne Chemical Co., Ltd. (Chengdu, China) and were used in accordance with the requirements received.

### 2.2. Liquid Phase Exfoliation of BNNS

The liquid phase processing process of boron nitride nanosheets is shown in Figure 1. In this experiment, the liquid stripping of h-BN was carried out using the hydrothermal method and water bath ultrasonic method to prepare the required BNNS. In this process, according to the previous literature [19,27,30], 1 g h-BN with a particle size of 10 μm was usually mixed with 1 g trisodium citrate dihydrate and added into the mixture of 100 mL isopropyl alcohol and water (1:3), and stirred to disperse evenly. Then, an ultrasonic probe (TL-650Y, Tianyi Instrument Co., Ltd., Cangzhou, China) was used to disperse the solution for 20 min so that the h-BN particles were evenly dispersed in the solution. The above-mixed solution was then treated with an ultrasonic water bath with a 40 kHz (200 W) ultrasonic two-dimensional material stripper (SCIENTZ-CHF-5A, Xinzhi Biotechnology Co., Ltd., Ningbo, China) for 6 h to obtain the dispersion solution. The obtained dispersion was removed and moved to a stainless-steel high-pressure reactor lined with polytetrafluoroethylene, where it was heat-treated in water at 180 °C for 6 h. After the heating was stopped, the dispersions were cooled in air, and then the dispersions obtained from hydrothermal treatment were taken out and treated with deionized water three times. After this treatment, the solids obtained were dispersed with deionized water, and then the dispersion was moved to a high-pressure reaction kettle. The water heat treatment was conducted at 180 °C for 3 h. After cooling, deionized water was used again for extraction and filtration to collect the solids obtained by extraction and filtration. Finally, 1 g of BNNS solid was collected after drying the solids at a 60 °C vacuum for 48 h.

### 2.3. Fabrication of Epoxy Composites

The preparation process of the composite materials is shown in Figure 1. This preparation mainly adopts the solution blending and curing process. Firstly, the BNNS powder was weighed and placed in a beaker. After 8 mL acetone was added, the BNNS powder was treated in a 40 kHz (200 W) ultrasonic two-dimensional material stripping water bath for 15 min and then stirred at 400 rpm for 30 min so that the BNNS were evenly dispersed in acetone and a uniform mixture was obtained.

In another beaker, we mixed the epoxy, hardener, and accelerator at a 100:80:1.6 weight ratio. G-POSS was then added to the epoxy mixture at room temperature to produce mixtures with different filling amounts (0.5, 1, 2, 3, 4, and 5 wt%) according to the weight of the epoxy, curing agent, and promoter. Then, on the basis of the weight of the epoxy resin, curing agent, and promoting agent, the required BNNSs dispersions were added to the above mixture with a 20 wt% filling amount to obtain BNNSs/G-POSS/epoxy complex. The compound was stirred at 500 rpm for 1 h at a constant speed to make the BNNSs, G-POSS, epoxy resin, and curing agent fully and evenly mixed while removing the small amount of acetone remaining in the mixture. After this operation, the compound was placed in a vacuum oven preheated at 60 °C and vacuumized and degassed at 60 °C for 1 h to remove most of the bubbles in the compound.

Then, the composite was slowly poured into a circular metal mold with a diameter of 30 mm and a thickness of 3 mm, and the vacuum was evacuated again until there was no obvious bubble overflow on the surface of the composite, and the air extraction was stopped. Then, the oven temperature was raised to 120 °C and cured for 2 h at 120 °C, and then the temperature was raised to 130 °C and cured for 2 h at 130 °C. Finally, the samples of BNNSs/G-POSS/epoxy resin composites with different fillings obtained by curing were removed after full cooling in air. After the samples were obtained, the solid surfaces were polished smooth with sandpaper and gold sprayed on the surfaces.

### 2.4. Characterizations

The microstructure and morphology of h-BN and BNNSs were characterized using scanning electron microscopy (SEM, 5 kV, ZEISS, Gemni400, Oberkochen, BW, Germany). Also, the microstructure of the cross-section obtained in the liquid nitrogen environment was characterized. The thermal conductivity of the composites was measured using the plane heat flow method (DRL-III thermal conductivity instrument, Xiangtan Instrument Company, Xiangtan, China) at room temperature of 25 °C. The dielectric constant and the tangent of the dielectric loss angle were measured in the frequency range of 10^1^–10^5^ Hz using a broadband dielectric impedance relaxation spectrometer Concept 50 (Novocontrol Technologies, Montabaur, Germany). The dynamic thermo-mechanical properties of the composites were measured using a dynamic thermo-mechanical analyzer (DMA850, Discovery, New York, NY, America) at a heating rate of 5 °C/min and a working frequency of 1 Hz. Thermogravimetric analysis (TGA) was performed using a synchronous thermal analyzer (Mettler-Toledo Technology Co., Ltd., Zurich, Switzerland) at a heating rate of 10 °C/min and nitrogen (20 mL/min) flow.

## 3. Results and Discussion

Figure 2a,b shows the SEM images of h-BN and BNNSs. The BNNSs were the liquid phase exfoliation of h-BN using the hydrothermal method and water bath ultrasonic method. Due to the high pressure and high temperature generated by the hydrothermal reaction and the effect of trisodium citrate added during mixing, the effect of h-BN between layers is weakened, and the distance between layers increases. Therefore, the thicker h-BN was separated into the singly thin BNNSs (Figure 2b). The SEM image of the cross-section of the pure epoxy sample is shown in Figure 2c. It can be seen that the cross-section is relatively flat without an obvious cavity formed by bubbles. The cross-sections of the G-POSS/epoxy composites did not change significantly after 5 wt% of G-POSS was filled into the epoxy resin. As shown in Figure 2d, the G-POSS molecule has good compatibility with epoxy, and its cross-section is also relatively smooth compared with that of the pure epoxy sample, forming a uniform structure without obvious agglomeration. This phenomenon also appears in the literature [28,31], which indicates that G-POSS molecules are distributed at the molecular level in epoxy resin. The cross-section SEM images of the BNNSs/epoxy composites containing 20 wt% of BNNSs and G-POSS/BNNSs/epoxy composites containing 20 wt% of BNNSs and 5 wt% of G-POSS are shown in Figure 2e,f. According to the SEM images, the BNNSs presented a single sheet in the epoxy resin, which was inlaid and distributed in different directions. Most BNNSs were overlapping with each other and were located on different planes, which formed a thermal pathway or three-dimensional thermal network in the resin. Since the particle size of the BNNSs is much larger than that of the G-POSS, it is difficult to observe the distribution of G-POSS in this structure.

In this study, h-BN with a particle size of 10 μm was selected as the filler material of the composite. This is because when selecting the h-BN with a small particle size, such as using 2 μm h-BN as the filler material, the number of particles in the filler process will be relatively large due to the small particle size of the raw material, which will lead to a relatively small distance between the particles compared with the 10 μm filler. Such a distance will cause greater gravity between the particles, and the displacement between the particles will become more difficult. Such a microscopic nature will improve the viscosity of the composite at the macro level. At the same time, when the particle size is small, the importance of the Brownian motion of the particle will increase, which will also lead to an increase in the viscosity of the composite. However, when h-BN with a particle size of 20 μm was used as raw material, the viscosity decreased slightly, and the h-BN with a particle size of 20 μm was difficult to peel. Therefore, 10 μm h-BN is an ideal filler material for this experiment.

Figure 3a,b shows the nanoholes in the G-POSS/epoxy composite with a 5 wt% of G-POSS at 500 nm and 1 μm scales. This indicates that with the increasing content of G-POSS, a certain number of nanoholes will be generated within the epoxy resin. A noticeable contrast in the structure can be observed when compared to the SEM image of the pure epoxy resin cross-section in Figure 2c.

Figure 3c–e depicts the EDX mapping images of the internal elements B, N, and Si within the G-POSS/BNNS/epoxy composite material. Given that the BNNSs are primarily composed of B and N elements, their abundant and uniform distribution within the epoxy resin can be inferred from the EDX mapping images of B and N. This distribution serves as crucial evidence for the mechanism behind the enhanced thermal conductivity. Similarly, considering that the characteristic elements in the molecular formula of G-POSS are Si, the distribution of G-POSS molecules within the epoxy can be determined through Si element EDX mapping. It is evident that the post-solvent blending processing leads to a uniform distribution of G-POSS molecules within the epoxy resin, thereby introducing numerous nanoscale micropores within it.

Figure 4a,b shows the thermal conductivity of BNNSs/G-POSS/epoxy composites changing with the contents of the BNNSs and G-POSS. Due to the excellent thermal conductivity of BNNSs, the thermal conductivity of the BNNSs/G-POSS/epoxy composites has been significantly improved with the increase in BNNSs contents.

In Figure 4a, the BNNSs were filled into epoxy to form BNNSs/epoxy composites. The thermal conductivity of the BNNSs/epoxy composites increases gradually with the increase in BNNSs filler concentration and reaches the maximum value of 0.9457 W·m^−1^·K^−1^ when the BNNSs filler concentration is 40 wt%. At this time, the thermal conductivity of the composite material is 511.47% of the thermal conductivity of the pure epoxy resin (0.1849 W·m^−1^·K^−1^), which has been greatly improved. The variation trend in the thermal conductivity of the composites can be analyzed using the theory of the thermal conduction pathway. According to the thermal conductivity pathway theory, when the filling amount of BNNSs is low, the number of BNNS particles is relatively small, and most BNNS particles exist in the epoxy resin in isolation, randomly embedded in the epoxy matrix, its periphery is covered by a large number of epoxy resin molecules, the high interface thermal resistance generated by the epoxy resin inhibits heat transfer. A dispersed system similar to a “sea-island” is formed, and the connection between each BNNS particle is not close. The number of thermal conduction channels in the composite material is small, the thermal conductivity is not good, and its thermal conductivity is relatively small. Therefore, when BNNSs filler concentration is 10 wt%, the thermal conductivity of the BNNSs/epoxy composite material is only 0.2495 W·m^−1^·K^−1^, compared with the thermal conductivity of pure epoxy resin, which increased by only 34.94% and did not improve by much. With the increase in the BNNS filling amounts, the average distance between BNNSs is shortened, which is different from the “sea-island” dispersed system with a low filling amount. The probability of BNNS contact in the composite increases, and then a dispersed system similar to “island-island” is formed, gradually forming a large number of heat conduction channels. When the filling amount reaches a certain critical value, a three-dimensional thermal conductivity network will be formed between BNNSs, which will greatly shorten the heat flow path along with the thermal resistance. Therefore, the thermal conductivity of the composite materials will increase with the increase in the filling amount of BNNSs [21]. When the concentration of the BNNS filler is 20 wt%, the thermal conductivity of the BNNSs/epoxy composites is 0.5496 W·m^−1^·K^−1^, which is 297.24% that of pure epoxy resin. When the concentration of the filler is 30 wt%, the thermal conductivity of the composite material reaches 0.8062 W·m^−1^·K^−1^, which is 436.02% that of pure epoxy resin. The thermal conductivity of the composite also reaches the maximum value when the filling amount reaches the highest 40 wt%. The variation trend of thermal conductivity is consistent with the expected gradient of BNNS filler in this experiment.

According to the above analysis, in order to enhance the thermal conductivity of G-POSS/epoxy composites, we chose BNNSs as the filler to enhance the thermal conductivity of the composites, consisting of the BNNSs/G-POSS/epoxy composite system. Figure 4b reflects the influence of the G-POSS filler amount on the thermal conductivity of BNNSs/G-POSS/epoxy composites when the filler amount of BNNSs is 20 wt%. Therefore, the changing trend of G-POSS filler concentration on thermal conductivity reflected in Figure 4b is not solely caused by G-POSS but by the simultaneous influence of the 20 wt% of BNNSs and G-POSS with different filler concentrations on the thermal conductivity of the epoxy resin. When the G-POSS filler concentration is 2 wt%, the thermal conductivity of the BNNS/G-POSS/epoxy composites is 0.3969 W·m^−1^·K^−1^, which is 214.66% that of pure epoxy resin (0.1849 W·m^−1^·K^−1^). The thermal conductivity of the BNNSs/epoxy composites decreased by 38.47% compared with 20 wt% of BNNSs filler. The same is true for BNNSs/G-POSS/epoxy composites with a G-POSS filler concentration of 5 wt%. It can be seen that after filling the G-POSS of six different quality gradients in the BNNSs/epoxy composite system containing 20 wt% of BNNSs, the thermal conductivity of the BNNSs/G-POSS/epoxy composite system formed by the G-POSS is not greatly changed. Compared with the BNNSs/epoxy composite system, it declined, but its thermal conductivity is still greatly improved compared to pure epoxy resin. The possible reason for the above phenomenon is that after the G-POSS is filled, with the increase in G-POSS content, the unique cage structure of the G-POSS will form a certain number of nanoholes in the composite material, which will hinder the flow of heat inside the composite material, thus reducing the thermal conductivity of the composite material. Simultaneously, due to the filling of the G-POSS, G-POSS molecules may appear to a certain degree of agglomeration in the interior of the composite material instead of being very evenly dispersed in the composite material, which leads to its failure to effectively cooperate with BNNSs to form a heat conduction pathway. The thermal conductivity of the BNNSs/G-POSS/epoxy composites is also decreased, which makes the thermal conductivity of the composites lower than that of the BNNSs/epoxy composites containing 20 wt% of BNNSs. However, the thermal conductivity of the composite is still about 110% higher than that of pure epoxy resin.

Apart from the high thermal conductivity performance of the composite material, its dielectric properties influence the insulation performance and are also very critical. Figure 5 shows the electrical resistivity changes of the G-POSS/BNNSs/epoxy composites with different G-POSS content under alternating electric fields at different frequencies. Because the G-POSS molecule is a cage structure formed by silicon-oxygen bonds, it has the same excellent electrical insulation performance as SiO_2_. BNNSs, meanwhile, also have excellent insulation performance. The two commons hindered the propagation path of the composite materials in electronics. This enabled the electrical resistivity of the composites at the low-frequency electric field under the action above 10^11^ Ω·cm in the high-frequency electric field under the action of above 10^10^ Ω·cm to meet the requirements of good insulation properties.

As for the dielectric constant of the sample, both G-POSS and BNNS fillers will affect the dielectric constant of the composite to a certain extent. As shown in Figure 6a, after filling 10%, 20%, 30%, and 40 wt% of BNNSs, respectively, the dielectric constant of the sample increases gradually compared with the pure epoxy. The dielectric constant of the BNNSs/epoxy composites reaches its maximum value of 6.05 at 1130 Hz when 40 wt% of BNNSs is filled. For the pure epoxy resin sample, its dielectric constant is 5.35 at 1130 Hz. It can be seen that the dielectric constant of BNNSs/epoxy composites with 40 wt% of BNNS filler is 13.08% higher than that of pure epoxy resin samples. That said, if BNNSs as fillers enhance the performance of the thermal conductivity of epoxy resin, it is bound to cause the rise of the epoxy resin’s dielectric constant. When the filler concentration is high, it is likely to lead to the complex dielectric constant rising too much beyond the boundary between the high and low dielectric constant of 10.0, greatly reducing the materials’ insulation performance. It cannot meet the requirements of both a low dielectric constant and high thermal conductivity of composite materials. According to [32,33], this is because BNNSs will form a large number of laminated systems in the composite material after the introduction of BNNSs, which will form a certain number of tiny capacitive structures in the epoxy resin and exist in the composite system as electrodes of capacitors. Between the stacked BNNS capacitors, the epoxy forms the dielectric of the capacitor. Sandwich polarization occurs under the action of a high-frequency alternating electric field, which increases the dielectric constant of the composite material. Since the introduction of BNNSs, the composite material’s internal interface will be increased, which makes interface polarization more likely to happen. The polarization increases and directly leads to a high dielectric constant. In addition, the composite material is also affected by the polarity of the BNNS itself, impurities, and internal bubbles, resulting in increased polarization, which also increases its dielectric constant. As for the dielectric loss of composite materials, filling BNNSs into the epoxy resin will lead to an increase in its dielectric loss, as shown in Figure 6f. After filling 20 wt% BNNSs into the epoxy resin, the dielectric loss of the composite will increase as a whole. It is at a relatively high level (tanδ > 0.01) under the action of a high-frequency electric field above 10^4^ Hz, which is the interlayer polarization and interface polarization loss caused by the introduction of BNNSs. Therefore, at the same time as filling BNNSs, it is necessary to fill another kind of filler that can reduce the dielectric constant and dielectric loss of epoxy resin, so as to reduce the dielectric constant and dielectric loss, which rises due to the filling of BNNSs. According to the literature [34], the filling of G-POSS can reduce the dielectric constant and dielectric loss of epoxy resin in a certain range. Therefore, G-POSS is selected as the second filler to reduce the dielectric constant and dielectric loss of composite materials in order to meet the performance requirements of low-dielectric insulating materials.

According to Figure 6b, after the G-POSS is filled into the epoxy resin, the dielectric constants of the G-POSS/epoxy composites with six different mass gradients are all lower than those of the pure epoxy resin to a certain extent. When filling G-POSS, the dielectric constant of G-POSS/epoxy composite decreases gradually with the increase in G-POSS content and reaches the lowest at the filler concentration of 2 wt% of G-POSS, that is, 4.88 at 1130 Hz, which is about 8.79% lower than the pure epoxy resin sample. When the content of BNNS is higher than 2 wt%, the dielectric constant of the composite begins to rise from the lowest point, and finally, when the content of G-POSS is 5 wt%, the dielectric constant of the composite is 5.24 at 1130 Hz, which is still lower than the dielectric constant of epoxy resin 5.35 at this frequency. It can be seen that G-POSS can reduce the dielectric constant of epoxy resin. Such performance is caused by the structure of G-POSS. G-POSS molecules are cage structures formed by silicon-oxygen bonds. These structures are surrounded by 8–12 active epoxy groups. It is precisely because of the existence of these groups that G-POSS molecules can be cured with epoxy molecules and curing agents and form covalent bonds, forming a close network crosslinked structure [35]. Thus, it is better dispersed in the epoxy. Unlike the stacked distribution of BNNSs in epoxy, G-POSS molecules are isolated in the epoxy matrix when the mass fraction is low, which reduces the probability of interface polarization, thus reducing the dielectric constant of the G-POSS/epoxy composites. With the increase in G-POSS molecular content, the network crosslinking structure formed by G-POSS and epoxy molecules will be reduced. Meanwhile, the aggregation of G-POSS molecules with high content will form overlapping interfaces, resulting in the occurrence of interface polarization and an increase in polarization such that the dielectric constant will rise to a certain extent. The silicon-oxygen structure of the G-POSS molecule itself makes it have a large dielectric constant, which will also increase the dielectric constant of the composite material. As shown in Figure 6d, after the G-POSS molecules were filled into the epoxy resin, the tangent of the dielectric loss angle of the composite material decreased. This is also caused by the structure of the G-POSS molecule. The strong structure formed by covalent bonds between epoxy groups around the G-POSS and epoxy molecules will prevent the generation of a conductive network. According to the above analysis, due to its special molecular structure, G-POSS can decrease the dielectric constant of epoxy resin in a certain range, and when its mass fraction is higher, the dielectric constant will gradually increase from a lower level. Therefore, 2 wt% of G-POSS can be selected as the ideal filler concentration to reduce the dielectric constant and dielectric loss of BNNSs/epoxy composites.

As shown in Figure 6c,e, based on selecting 20 wt% of BNNSs as the improvement in thermal conductivity of composite materials, G-POSS molecules with different mass gradients are filled to neutralize the increase in the dielectric constant and dielectric loss of composite materials caused by BNNSs. According to Figure 6f, the dielectric constant of BNNSs/epoxy composites containing 20 wt% of BNNS is 5.57 at 1130 Hz, which is higher than the dielectric constant of G-POSS/BNNSs/epoxy composites. This indicates that after filling the G-POSS molecules, the dielectric constant of the composite materials does decrease, and the changing law is consistent with the above G-POSS/epoxy composite system. It all decreases first when the content is less than or equal to 2 wt% and then rises after the content is more than 2 wt%. Furthermore, when the G-POSS content is 2 wt%, the dielectric constant of the composite is 5.28 at 1130 Hz, which is the minimum value among all gradients. This is a 5.21% decrease compared with the BNNSs/epoxy composite system. At the same time, this value is slightly lower than the dielectric constant of 5.35 of pure epoxy resin at this frequency. In conclusion, G-POSS filler can greatly improve the dielectric properties of composite materials so that its thermal conductivity is improved and its dielectric properties are also better compared with that of pure epoxy resin materials so as to meet the performance requirements of insulating materials with low dielectric and high thermal conductivity.

Figure 7a,b are the curves of the loss modulus and energy storage modulus of the epoxy and G-POSS/BNNSs/epoxy composites as a function of temperature. According to the tanδ curve, the glass transition temperature (T_g_) of the composites formed by filling the epoxy resin with 5 wt% of G-POSS nanofiller and 20 wt% of BNNSs is 127.94 °C, which is higher than that of the pure epoxy resin (126.26 °C). This may be because G-POSS disperses evenly in epoxy resin after filling G-POSS nanofillers. The special cage structure of G-POSS will interact with BNNSs to inhibit the movement of the epoxy molecular chain to a certain extent and increase the energy required for its movement, thus increasing the glass transition temperature. As for the energy storage modulus of composites, the energy storage modulus of the G-POSS/BNNSs/epoxy composites increased significantly after the addition of BNNSs, indicating that BNNSs and G-POSS jointly restrict the migration of the epoxy molecular chain [36,37].

Figure 7c shows the TGA curve of the composite as a function of temperature. From the analysis of the figures, it can be known that the thermal weight loss of the composites mainly occurs within the temperature range of 300~500 °C. When the thermal loss weight is 5% of the total mass, it is the initial decomposition temperature of the composite material. According to the curve, pure epoxy resin’s initial decomposition temperature is 342.8 °C. The initial decomposition temperature of G-POSS/BNNSs/epoxy is 346.3 °C, which is about 1.02% higher than that of pure epoxy. In the meantime, the initial decomposition temperature of G-POSS/epoxy composites containing 2 wt% and 5 wt% of G-POSS is also higher than that of pure epoxy. These results indicate that the addition of G-POSS and BNNSs can improve the initial decomposition temperature of epoxy resin.

According to the calculation formula of the heat-resistance index temperature:(1)$$T_{HRI}=0.49\times [T_{5}+0.6\times (T_{30}-T_{5})]$$

*T*_5_ is the thermal decomposition temperature when the mass loss of the composite is 5% of the total mass, while *T*_30_ is the thermal decomposition temperature at the time the mass loss is 30% [17]. The thermal index temperature of G-POSS/BNNSs/epoxy composites is 190.5 °C, and that of pure epoxy is 183.1 °C, which indicates that BNNSs and G-POSS are able to enhance the thermal stability of epoxy resins.

## 4. Conclusions

In this study, by modifying the epoxy resin matrix with G-POSS and BNNSs, the G-POSS/BNNSs/epoxy composite containing G-POSS nano-filler with different mass gradients was prepared, respectively. According to the performance test results, the thermal conductivity of the composite can reach 0.3969 W·m^−1^·K^−1^ when the content of BNNSs and G-POSS is 20 wt% and 2 wt%, respectively. This is because BNNSs create a continuous network of heat conductions inside the epoxy resin, increasing the rate of heat transfer. Meanwhile, the dielectric constant of the G-POSS/BNNSs/epoxy composites is 5.33, and the tangent of dielectric loss angle tanδ value is less than 0.01. This is because G-POSS reacts with epoxy resin to form a strong network structure, which reduces the occurrence of interfacial polarization and also hinders the transfer of electrons inside the composite material, improving the dielectric properties of the composite material. Not only that, but the composite material also has a higher resistivity. Due to the addition of BNNSs and G-POSS, both the dynamic mechanical properties and thermal stability of composites are improved. In conclusion, G-POSS/BNNSs/epoxy composite material shows excellent thermal conductivity and dielectric properties and can be well used in high-pressure insulation equipment with increasingly serious heat dissipation problems.

## Figures and Tables

**Figure 1 nanomaterials-14-00205-f001:**
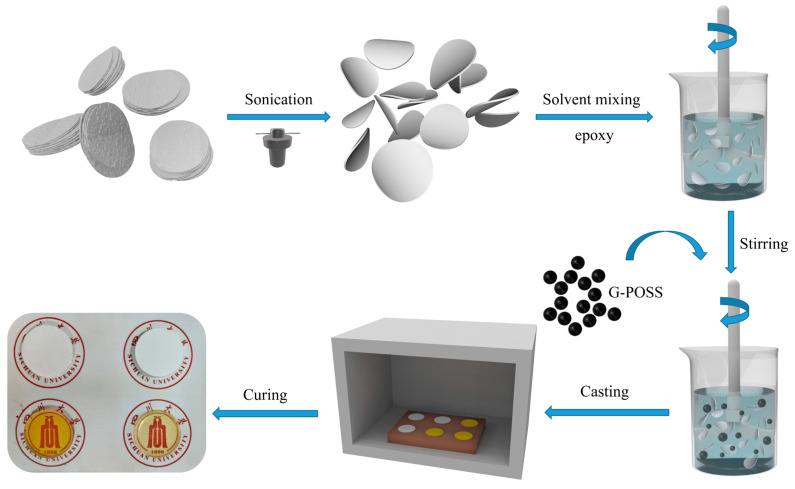
Fabrication of G−POSS/BNNSs/epoxy composites.

**Figure 2 nanomaterials-14-00205-f002:**
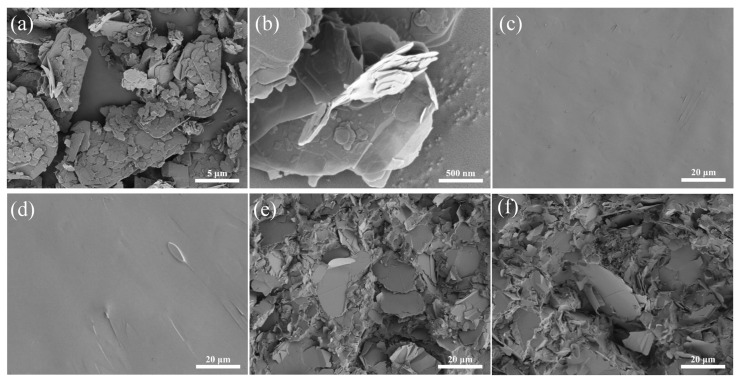
SEM images of the (**a**) h−BN and (**b**) BNNSs. Typical SEM images of the cryogenically fractured surface morphologies of the (**c**) epoxy, (**d**) G−POSS/epoxy composite, (**e**) BNNSs/epoxy composite, and (**f**) G−POSS/BNNSs/epoxy composite, respectively. All the composites were brittle and fractured in liquid nitrogen, and the obtained sections were sprayed with gold and characterized using SEM.

**Figure 3 nanomaterials-14-00205-f003:**
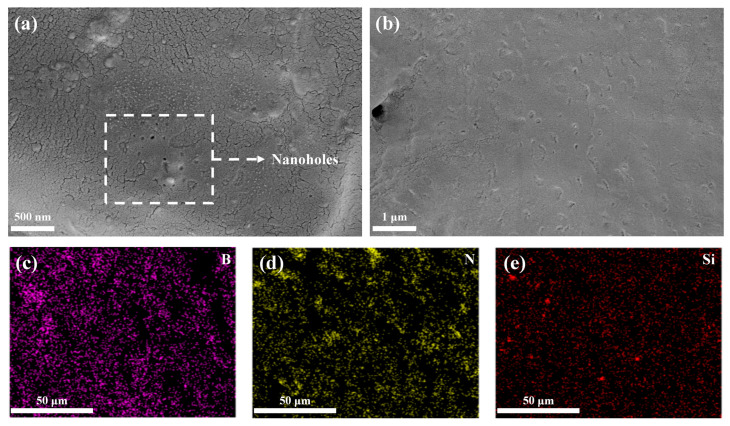
SEM images of the nanoholes in the G−POSS/epoxy composite with 5 wt% of G−POSS at the (**a**) 500 nm and (**b**) 1 μm scales and (**c**–**e**) energy dispersive X−ray mapping images of the B, N, and Si elements of the G−POSS/BNNS/epoxy composite.

**Figure 4 nanomaterials-14-00205-f004:**
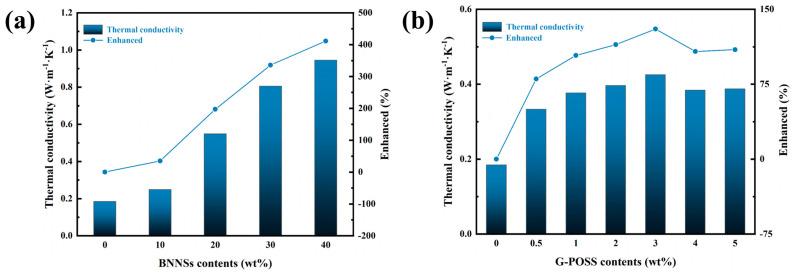
Thermally conductive properties of the composites. (**a**) Thermal conductivity of the BNNSs/epoxy composites varies with the BNNSs contents. (**b**) Thermal conductivity of the BNNSs/G−POSS/epoxy composites (fixed BNNS loading at 20 wt%) varies with the G−POSS contents.

**Figure 5 nanomaterials-14-00205-f005:**
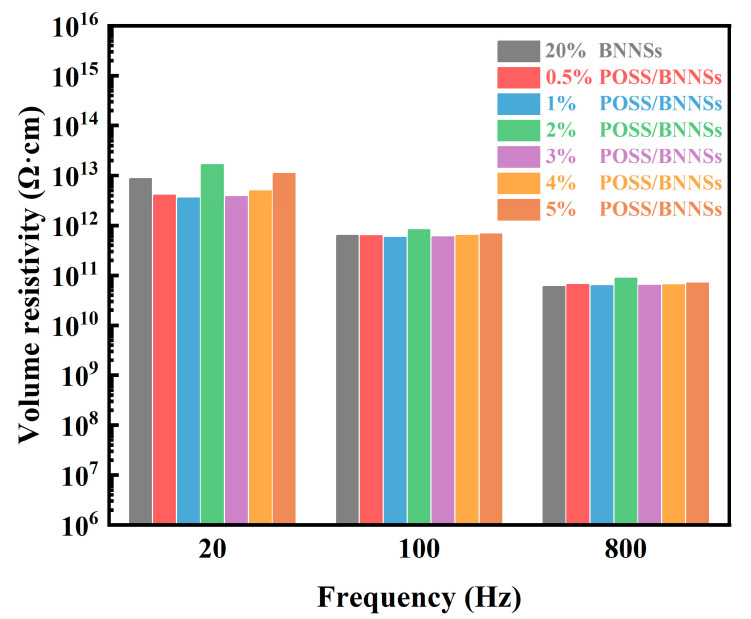
Volume resistivity of the G−POSS/BNNSs/epoxy composites at different frequencies.

**Figure 6 nanomaterials-14-00205-f006:**
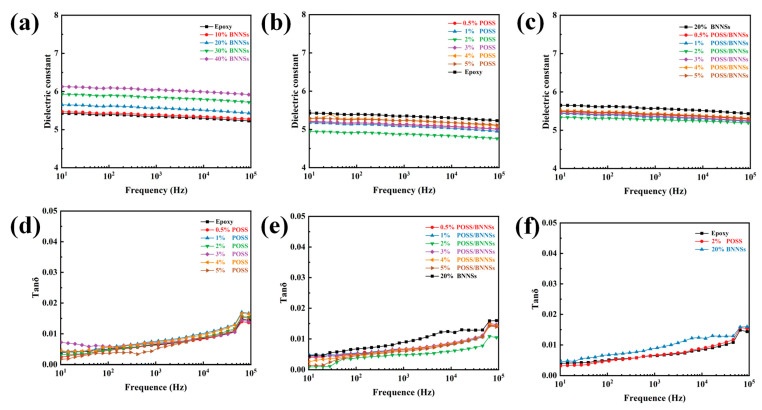
Dielectric properties of the BNNSs/epoxy, G−POSS/epoxy, and G−POSS/BNNSs/epoxy composite materials. (**a**–**c**) The dielectric constant of BNNSs/epoxy, G−POSS/epoxy, and G−POSS/BNNSs/epoxy composite materials, respectively. (**d**,**e**) The dielectric loss of G−POSS/epoxy and G−POSS/BNNSs/epoxy composite materials. (**f**) Comparison of dielectric losses of epoxy, G−POSS/epoxy, and BNNS/epoxy.

**Figure 7 nanomaterials-14-00205-f007:**
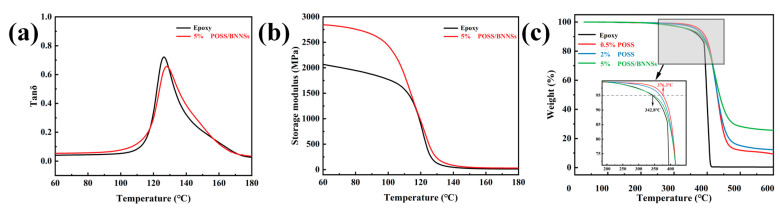
Dynamic mechanical properties and thermal stability of the G−POSS/BNNSs/epoxy composites. (**a**,**b**) Loss factor and storage modulus curves of epoxy and G−POSS/BNNSs/epoxy composites. (**c**) TGA curves of epoxy, G−POSS/epoxy composites, and G−POSS/BNNSs/epoxy composites.

## Data Availability

The data presented in this study are available on request from the corresponding author.

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
