# Peer review of "Significantly Improve the Thermal Conductivity and Dielectric Performance of Epoxy Composite by Introducing Boron Nitride and POSS"

_nanomaterials, 2024, doi:10.3390/nano14020205_

Round 1

Reviewer 1 Report

Comments and Suggestions for Authors

The authors measured thermal conductivity and dielectric constant of the epoxy composites. It was found that introducing boron nitride or G-POSS into epoxy resin enhanced thermal conductivity. There have been a lot of works about the thermal conductivities of composites. The results the authors obtained in this work are useful for a lot of researchers. However, there are some concerns the authors should address before accepting this manuscript in nanomaterials. If the authors revised the manuscript appropriately, this study would meet the criteria for the publication.

Comment list

Comment 1: When increasing the amount of G-POSS, a certain number of nanoholes are generated. Do you have the evidence for that? It is important to discuss the relationship between thermal conductivity and the amount of G-POSS.

Comment 2: In Figure 4, the authors discussed the resistivities of epoxy composites with various volume fractions of POSS/BNNSs. Why did the sample with 2% POSS/BNNSs exhibit higher resistivity?

Comment 3: Do you have the EDX mapping image? I would like to know the distribution of boron nitride or G-POSS in epoxy resin.

Comment 4: In the introduction, the authors mentioned the thermal management performance. This is very hot topic for efficiently reusing thermal energy. Therefore, the authors cannot ignore important works about thermal management. For example, there are some important works: thermal switch (Nano Lett. 22, 6105-6111 (2022).), thermal barrier (Nano Lett. 15, 6809-6814 (2015).). The authors should cite these recommended works at least. This will improve the quality of this work.

Author Response

Dear reviewer,

Thank you very much for your helpful review and constructive suggestions regarding our manuscript “Significantly Improve the Thermal Conductivity and Dielectric Performance of Epoxy Composite by Introducing Boron Nitride and POSS” (Manuscript ID:2786065 ). We have carefully revised the manuscript according to the comments and have also provided a point-by-point response to the comments below.

The authors measured thermal conductivity and dielectric constant of the epoxy composites. It was found that introducing boron nitride or G-POSS into epoxy resin enhanced thermal conductivity. There have been a lot of works about the thermal conductivities of composites. The results the authors obtained in this work are useful for a lot of researchers. However, there are some concerns the authors should address before accepting this manuscript in nanomaterials. If the authors revised the manuscript appropriately, this study would meet the criteria for the publication.

Authors reply. Thank you very much for your recommendation.

Comment 1: When increasing the amount of G-POSS, a certain number of nanoholes are generated. Do you have the evidence for that? It is important to discuss the relationship between thermal conductivity and the amount of G-POSS.

Authors reply. Thanks for your comments. The SEM image shown in Figure (a) illustrates the fractured surface of G-POSS/Epoxy composite containing 5 wt% of G-POSS after undergoing cryogenic fracturing. At a microscopic scale of 500 nm, a significant presence of nanoscale-sized pores can be distinctly observed within the region highlighted by the box on the fracture surface. Figure (b) presents a SEM image at a larger scale, where, as depicted, numerous nanoscale pores are distributed across the surface of the composite containing 5 wt% of G-POSS, observable at a scale of 1 μm.

Meanwhile, we have included relevant images and discussions to demonstrate that with the increase in G-POSS content, a certain number of nanoholes form within the epoxy resin. The details can be found in Line 202, Page 5 in the revised manuscript. The newly added images are labeled as Figure 3a and Figure 3b in the revised manuscript.

Comment 2: In Figure 4, the authors discussed the resistivities of epoxy composites with various volume fractions of POSS/BNNSs. Why did the sample with 2% POSS/BNNSs exhibit higher resistivity?

Authors reply. Thanks for your comments. Actually, the POSS/BNNS/epoxy composites at different POSS loading have similar resistivity values. We repeatedly measured the properties of the POSS/BNNS/Epoxy composite at 2% filler contents. The higher resistivity may be attributed to the good dispersion of POSS. Higher POSS loading may resulting in aggregation, leading to deteriorate of the insulation properties.

Comment 3: Do you have the EDX mapping image? I would like to know the distribution of boron nitride or G-POSS in epoxy resin.

Authors reply. Thanks for your comments. The EDX mapping images shown in Figure (c) depict the distribution of three elements, B, N, and Si, within the G-POSS/BNNS/Epoxy composite. The EDX mapping of B and N distinctly illustrates the abundant and uniform distribution of BNNSs within the composite material. Similarly, due to the significant presence of Si in G-POSS molecules while epoxy resin is primarily composed of C and O elements, the EDX mapping of Si adequately illustrates the distribution of G-POSS molecules within the composite material.

Also, the discussion regarding the EDX mapping of both BNNS and G-POSS can be found in Line 207, Page 5 in the revised manuscript. The EDX mapping images for elements B, N, and Si are denoted as Figures 3c to 3e in the revised manuscript.

Comment 4: In the introduction, the authors mentioned the thermal management performance. This is very hot topic for efficiently reusing thermal energy. Therefore, the authors cannot ignore important works about thermal management. For example, there are some important works: thermal switch (Nano Lett. 22, 6105-6111 (2022).), thermal barrier (Nano Lett. 15, 6809-6814 (2015).). The authors should cite these recommended works at least. This will improve the quality of this work.

Authors reply. We are appreciate for your comments and suggestion. The mentioned papers have been reviewed and cited as Ref.[10] and Ref.[11] in the revised manuscript.

Reviewer 2 Report

Comments and Suggestions for Authors

The paper presents the study about improve the thermal conductivity and dielectric performance of epoxy composite by introducing boron nitride and POSS. Authors propose a strategy to improve the performance of epoxy composites by employing boron nitride nanosheets (BNNSs) and γ-glycidyl ether oxypropyl sesimoxane (G-POSS). Obtained epoxy composites exhibit high electrical resistivity (10 power 13 Ω·cm) and low dielectric loss (<0.01) due to the ultra-low dielectric constants. Obtained material has high thermal conductivity what is higher than that of pure epoxy resin.

Dear author, thank you very much for interesting paper about the improving the epoxy composite, used in electric power devices. I put some comments and question.

Comments:

1. The introduction is well written.

2. Epoxy composite is used mostly as material in case of high voltage insulators of overhead power transmission lines. Please explain why we should expect high thermal conductivity in that case.

3. You mentioned about dielectric losses (tan(delta)). Please explain some physical fundamentals of this property.

4. You mentioned about dielectric constant. There are some cases where we want high OR small dielectric constant. Please tell what case do you investigate.

5. Please say something about physical fundamentals of dielectric constant.

6. You mentioned about volume resistivity. Please say something about physical fundamentals of this property.

7. One of the more important property of materials, used in electric power engineering, is electric breakdown voltage. Please say, that your next investigations will be involved in this topic.

8. Generally conclusions: the paper presents very value results. Please complete some information, I asked.

Author Response

Dear reviewer,

Thank you very much for your helpful review and constructive suggestions regarding our manuscript “Significantly Improve the Thermal Conductivity and Dielectric Performance of Epoxy Composite by Introducing Boron Nitride and POSS” (Manuscript ID:2786065 ). We have carefully revised the manuscript according to the comments and have also provided a point-by-point response to the comments below.

The paper presents the study about improve the thermal conductivity and dielectric performance of epoxy composite by introducing boron nitride and POSS. Authors propose a strategy to improve the performance of epoxy composites by employing boron nitride nanosheets (BNNSs) and γ-glycidyl ether oxypropyl sesimoxane (G-POSS). Obtained epoxy composites exhibit high electrical resistivity (1013 Ω·cm) and low dielectric loss (<0.01) due to the ultra-low dielectric constants. Obtained material has high thermal conductivity what is higher than that of pure epoxy resin. Dear author, thank you very much for interesting paper about the improving the epoxy composite, used in electric power devices. I put some comments and question.

Q1. The introduction is well written.

Authors reply. Thank you very much for your comments.

Q2. Epoxy composite is used mostly as material in case of high voltage insulators of overhead power transmission lines. Please explain why we should expect high thermal conductivity in that case.

Authors reply. Thank you very much for your comments. Yes, high voltage insulators are one of the common application scenarios of epoxy resin. In addition, in some coil equipment such as switch cabinets, reactors, transformers, epoxy resin is also often used as insulation materials. These devices are easy to generate a lot of heat during operation, therefore, it is necessary to increase the thermal conductivity of epoxy resin to reduce the hot spot temperature of the equipment.

Q3. You mentioned about dielectric losses (tan(delta)). Please explain some physical fundamentals of this property.

Authors reply. Thank you very much for your comments. Dielectric loss refers to the phenomenon that the dielectric itself heats up due to the consumption of part of the electric energy in the alternating electric field. The reason is that the dielectric contains conductive carrier, which generates conductive current under the action of applied electric field, consumes part of the electric energy, and turns into heat energy.

Also, we have added related discussion on the importance of the dielectric loss and dielectric constants of materials in Line 49-52, Page 2 in the revised manuscript.

Q4. You mentioned about dielectric constant. There are some cases where we want high OR small dielectric constant. Please tell what case do you investigate.

Authors reply. Thank you very much for your comments. In this work, we want to get a low dielectric constant composite material.Therefore, we choose POSS, the hollow structure of POSS is conducive to reducing the dielectric constant and dielectric loss of the composite.

Q5. Please say something about physical fundamentals of dielectric constant.

Authors reply. Thank you very much for your comments. Dielectric constant refers to the ability of a material to retain electric charge, and the dielectric constant increases with the increase of molecular dipole moment and polarizability.

Q6. You mentioned about volume resistivity. Please say something about physical fundamentals of this property.

Authors reply. Thank you very much for your comments. Volume resistivity is the impedance of a material per unit volume to the current, which is used to characterize the electrical properties of the material. Generally, the higher the volume resistivity, the higher the efficiency of the material used as an electrical insulation component.

Q7. One of the more important property of materials, used in electric power engineering, is electric breakdown voltage. Please say, that your next investigations will be involved in this topic.

Authors reply. Thank you very much for your comments. Yes, it is indeed that excellent electrical properties are essential for the application of materials in high-voltage equipment. Our next work will focus on the electrical insulation properties of epoxy composites, including breakdown characteristics, charge transfer, traps, etc.

Q8. Generally conclusions: the paper presents very value results. Please complete some information, I asked.

Authors reply. Thank you very much for your recognition of our work.

Round 2

Reviewer 1 Report

Comments and Suggestions for Authors

Everything was cleared. This is worth publishing.